# User abnormal behavior recommendation via multilayer network

Chengyun Song[1], Weiyi Liu[2]*, Zhining Liu[3], Xiaoyang Liu[1]

**1** School of Computer Science and Engineering, Chongqing University of Technology, Chongqing, China, **2** JD Urban Computing Business Unit, Chengdu, China, **3** School of Information and Communication Engineering, University of Electronic Science and Technology of China, Chengdu, China

* jrliuweiyi@jd.com

## Abstract

With the growing popularity of online services such as online banking and online shopping, one of the essential research topics is how to build a privacy-preserving user abnormal behavior recommendation system. However, a machine-learning based system may present a dilemma. On one aspect, such system requires large volume of features to pre-train the model, but on another aspect, it is challenging to design usable features without looking to plaintext private data. In this paper, we propose an unorthodox approach involving graph analysis to resolve this dilemma and build a novel private-preserving recommendation system under a multilayer network framework. In experiments, we use a large, state-of-the-art dataset (containing more than 40,000 nodes and 43 million encrypted features) to evaluate the recommendation ability of our system on abnormal user behavior, yielding an overall precision rate of around 0.9, a recall rate of 1.0, and an F1-score of around 0.94. Also, we have also reported a linear time complexity for our system. Last, we deploy our system on the "Wenjuanxing" crowd-sourced system and "Amazon Mechanical Turk" for other users to evaluate in all aspects. The result shows that almost all feedbacks have achieved up to 85% satisfaction.

**Data Availability Statement:** The data used in this paper is from a third party (alibba company), and is freely available for download from https://tianchi.aliyun.com/datalab/dataSet.htm?spm=5176.100073.888.2o7.40bc1022nJtK6m&id=20. The authors did not have any special access privileges that others would not have.

## Introduction

With the recent growth of web activity such as online shopping and banking, it is becoming more and more necessary to detect abnormal user behavior on the web. For example, we can safeguard privacy for real users by distinguishing instances of identity theft.

In general, given the popularity of a wide variety of online services, machine learning (ML) algorithms including Support Vector Machine (SVM), decision trees, and deep neural networks have been applied to help operators maintain user privacy (See [1–3] for a review). Typically, after feeding enough features into a ML algorithm, a well-trained model can create an accurate portrait of the user, which can be used to protect users on the web. However, detecting abnormal user behavior can present a dilemma. For one, security operators must design accurate features that characterize user behavior. However, such features may require "peeking" at detailed user information such as Internet Protocol (IP) addresses, cookies, and

**Funding:** This work was supported by National Natural Science Foundation of China (No.41804112), Scientific Research Foundation of Chongqing University of Technology (No.2019ZD07) and Chongqing Research Program of Basic Research and Frontier Technology (No. cstc2018jcyjAX0287) to CS. Additionally, JD Urban computing business unit provided support for this study in the form of salary for WL. The specific role of this author is articulated in the 'author contributions' section.

**Competing interests:** The corresponding author (Weiyi Liu) is an employee in JD Urban computing business unit. The company only provided support in the form of salaries for Weiyi Liu but did not have any additional role in the study design, data collection and analysis, decision to publish, or preparation of the manuscript. This does not alter our adherence to PLOS ONE policies on sharing data and materials.

locations, which generally violates user privacy agreements. In addition, standard ML models require a multitude of features to build a well-trained model, but it is difficult to work with or collect such large amounts of data. Insufficient amounts of data can lead to over-fitting in resultant models.

Recently, supervised ML methods [4–7] have been proposed that attempt to preserve user privacy. For example, some methods use only encrypted features (instead of plaintext/unencrypted features) to train the model, but this use of encrypted data increases computational complexity [4]; others attempt to minimize the feature exposure possibility by decentralizing features over different distributed machines—but the host still has access to the whole set of features.

In this paper, we use advanced graph theory embedding algorithms to address the above problems. That is, based on the relationships among different behaviors for each user, we construct a topological structure (a graph) which uniquely depicts the behavior patterns for each user. Hence, unlike ML algorithms which must first train a model, graph-based methods naturally capture dependencies among the various features to yield a topologically-based portrait of the user. In addition, we also use a multilayer network [8, 9] to depict behavior patterns of the user over time("multilayer" represents different timestamps of the behavior patterns of a user on the web).

In conclusion, by using multilayer network analysis algorithms, we here develop a privacy-preserving user abnormal behavior recommendation system. Below are five major contributions:

1. **Privacy preservation**: As encrypted features are sufficient to construct a multilayer network, our system does not need access to unencrypted features and thus cannot leak information to host machines.

2. **Efficient feature use given limited features**: Our system can work with only one feature to construct a multilayer network based on user timestamps, in contrast to ML based algorithms which generally are difficult to train given insufficient features. However, the more features provided to our system, the more complete the user portrait it can build, and the more accurate abnormal behavior recommendations it can provide.

3. **Unsupervised learning**: As the multilayer network of a user captures user portraits automatically, we need no label information to train a model in the first place. Unsupervised network analysis algorithms such as community detection, sub-graph pattern matching, network embedding, and the like can also be applied to recommend abnormal behavior for the user automatically.

4. **Efficient performance**: Since our algorithms are quite efficient (see Section 16), and the proposed system works for each user, it is efficient to recognize abnormal behavior(s) for each user in the real time.

5. **A Practical System**: We have built a system that to visualize the results, demonstrate how to use the user's devices and the encrypted features to build the multilayer network, and recommend the suspicious activities. The system is available at https://github.com/Liu-WeiYi/Private_Preserving_Outlier_Behavior_Detection.

## Related work

Techniques based on ML [10, 11] and on graph analysis [12] algorithms have been developed to detect abnormalities. Generally speaking, these papers focus on a single problem: given a set

of features, how to construct a model which detects outliers in the dataset efficiently and accurately. In addition, to make input features more robust to non-linear noise, techniques such as denoising autoencoders [13, 14], maximum correntropy autoencoders [15], robust deep autoencoders [16], and malware detection on deep neural networks [17] have been proposed.

However, as pointed out by Shmatikov et al. [18], ML models remember too much. This property of ML algorithms may result in leaks of detailed information about the training dataset to malicious ML library providers. Hence, starting with Goldwasser et al. [4], researchers have begun to take privacy preservation into account by discussing how to leverage encrypted features during training [5, 6, 19–21]. Another way to prevent information leakage is to use distributed processing. For example, Shokri et al. [22] proposed a distributed training technique based on selective stochastic gradient descent, Sunil et al. [23] developed a novel attribute-wise noise addition scheme that preserves data privacy under guarantees of differential privacy, and Xie et al. [7] proposed a privacy-preserving proximal gradient algorithm which asynchronously updates models of the learning tasks. See [7] for a detailed review.

Unlike typical solutions which focus on constructing a more accurate or more privacy-preserving model to depict the data and capture outliers, we propose an unorthodox approach that focuses only on a specific user. As pointed out in the survey [12], graph-based approaches to anomaly detection have four advantages: the inter-dependent nature of the data, a powerful representation, the relational nature of problem domains, and robust machinery. In the proposed approach, we also use graphs to capture user behavior. In addition, we introduce a multilayer network [8, 9] to capture user behavior across time. As a multilayer network is a set of layers where each layer represents a type of relationship among nodes, it is natural to leverage layers to represent the user behavior for the current timestamp (or group of timestamps).

Generally speaking, there are two ways to extract useful information from the network: multilayer network community detection derived from pure topology analysis, and network embedding to find an appropriate vector space onto which to project nodes. For the former, many approaches use multilayer network community detection [24–27]. Among these, multi-slice modularity-based methods [28, 29] have proved efficient and accurate in detecting non-overlapping communities in (weighted) multilayer networks (reviewed in [30]). For the latter, by treating a stream of short random walks on a graph as a document, these methods project nodes onto a continuous vector space which serves as a social representative of the graph's nodes [31]. There continue to be approaches based on network embedding [31–37]: see [38] for a review. Liu et al. [39] has proposed three methods—network aggregation, results aggregation, and layer co-analysis—for multilayer network scenarios. In this paper, we use layer co-analysis to project the multilayer network onto a vector space.

## Proposed recommendation system

### One possible use case

We use the following example to describe our private-preserving user abnormal behavior recommendation system. Part of protecting online accounts is carefully preserving user privacy. That is, without violating user privacy (or using encrypted features instead), a privacy-preserving system must distinguish from the user behavior whether the account is currently being used by the real owner or an imposter. Fig 1 is an example.

### Recommendation system architecture

Fig 2 demonstrates the workflow of the proposed privacy-preserving abnormal user behavior recommendation system. In our system, there are four steps to recommend and visualize user's anomaly:

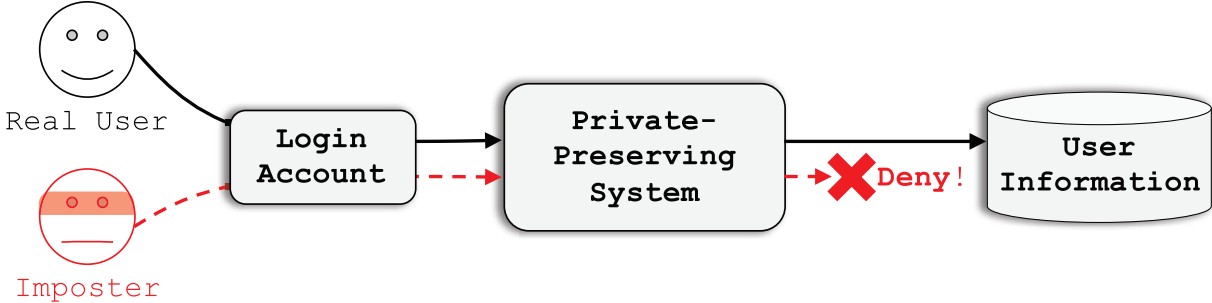

**Fig 1. A use case example.** A privacy-preserving system should distinguishes real users from imposters even though both are using the same login account, and forbids the imposter from accessing sensitive user data.

1. **Multilayer network construction**: This module captures user behavior across time (Alg. 1) and constructs a multilayer network based on these timeslot groups (Alg. 2), where each layer depicts the user behavior corresponding to a specific timeslot group. See Section for details.

2. **Multilayer network analysis**: This module detects abnormal users by using unsupervised graph analysis algorithms from two aspects: weighted multilayer network community detection based on the pure topology-based analysis (Alg. 3), and state-of-the-art multilayer network embedding (Alg. 4). See Section 12 for details.

3. **Outlier recommendation**: This module calculates node scores based on community and cluster results from the previous module (Alg. 5), and recommends nodes given the corresponding scores from small to large. See Section 10 for details.

4. **Visualization system**: This module builds a visualization system. See Section 16 for details.

## Multilayer network construction

**Multilayer network.** A multilayer network $MN = \{G^1, G^2, \ldots, G^l\}$ is a combination of layers, where each layer $G^l = (V^l, E^l)$ ($V^l$: node-set on layer $l$, $E^l$: edge-set on layer $l$) represents a particular type of relationship among nodes. In order to capture user behavior across time, nodes within the multilayer network can be devices belonging to a given user, the edge weights can be the similarity among different devices, and the layers can represent topological information among nodes with different timestamps ($t_1, t_2, \ldots$).

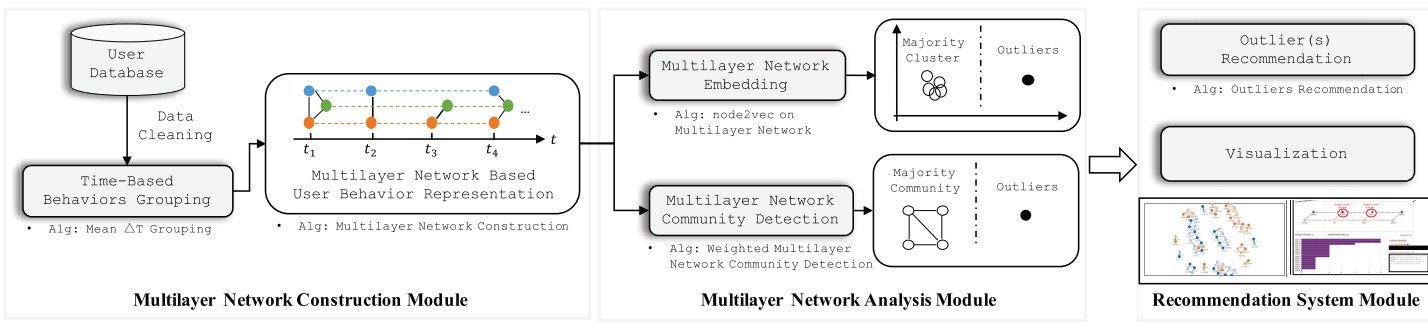

**Fig 2. Workflow for proposed system.**

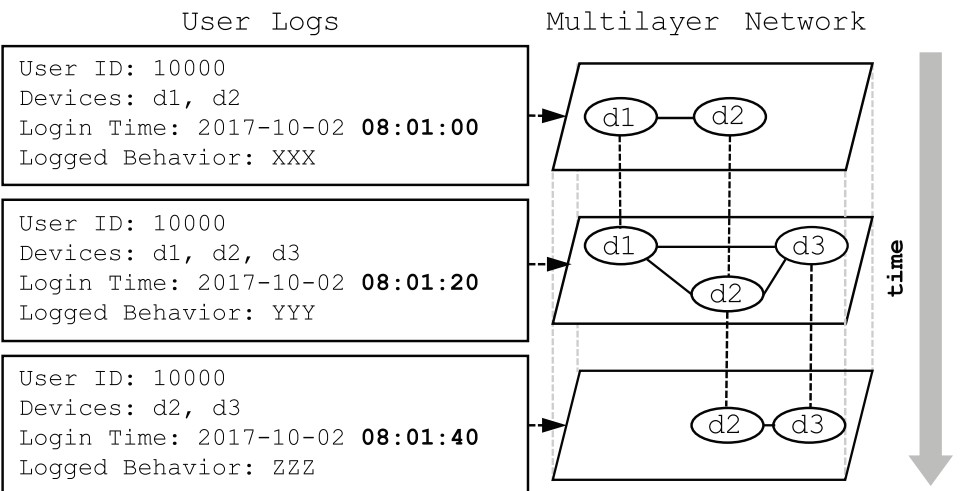

**Fig 3. Multilayer network based on user behavior over time.**

**Example.** Consider Fig 3. For user A0001, user logs register related devices and record behavior in the current timeslot group, and the multilayer network shows device relationships based on the behavior logged for the various timestamps. Dashed edges across layers link the same devices across different timestamps. Thus, when creating the multilayer network based on user behavior over time, there are two important processes: determining the layers, and determining the edge weights between the devices for each layer.

**Determining layers.** It is not practical to simply create a layer for each timestamp, as this would yield layers that contain redundant information. One way to solve this problem is to first group timestamps, and then create a layer corresponding to the current timestamp group. Alg. 1 shows the process by which timestamps are grouped based on the mean time interval ($\Delta T$), where the input is $All\_Time\_List = [t_1, t_2, \ldots, t_l]$ which contains all timestamps for the current user ($l$ is the total number of timestamps for the current user) and the output is $Time\_Groups = [g_{t_1}, g_{t_2}, \ldots, g_{t_m}]$ which contains each timeslot group $g_{t_1} = [t_1, t_2, \ldots]$ ($m$ is the total number of groups; $m \leq l$).

**Algorithm 1**: Mean $\Delta T$ Grouping

```
Input: All_Time_List
Output: Time_Groups
1 Time_Groups = [];
2 // 1. Calculate ΔT;
3 Interval_T = [];
4 for t₁_idx ≤ len(All_Time_List) - 1 do
5    t₂_idx = t₁_idx + 1;
6    t₁ = All_Time_List[t₁_idx];
7    t₂ = All_Time_List[t₂_idx];
8    Interval_T.append(t₂ - t₁);
9 end
10 ΔT = mean(Interval_T);
11 // 2. Group times by ΔT;
12 for t₁, t₂ ∈ All_Time_List do
13    for t₂ - t₁ ≤ ΔT then
14       Time_Groups[-1].extend(t₁, t₂);
15    end
16    else
17       Time_Groups.append(t₁, t₂);
```

```
18    end
19  end
20 return Time_Groups
```

**Determining edge weights.** After obtaining *Time_Grou −ps* for the current user, the weights between the different devices in the current timeslot group $W(d_1, d_2) \in [0, 1]$ are calculated using Eq 1. $B(\cdot)$ contains all the behavior features (such as scrambled IP address and encrypted queried keywords) of the current device in the current timeslot group.

$$W(d_1, d_2) = \frac{|B(d_1) \cap B(d_2)|}{|B(d_1) \cup B(d_2)|} \qquad (1)$$

Alg. 2 details the construction of the multilayer network based on *Time_Groups*, where the inputs are *Time_Groups* and *User_Logs* and the output is a weighted multilayer network *MN* for the corresponding user. Note that Eq 1 requires at least one feature to calculate the similarity between two devices (Produces binary output if fed only one feature); however, the more features obtained for a device, the more closely the resulting weights approximate the similarities between the corresponding devices.

**Algorithm 2**: Multilayer Network Construction

```
Input: Time_Groups, User_Logs
Output: MN
1 Initialize a multilayer network MN;
2 for g_{t_i} ∈ Time_Groups do
3   Initialize a graph G in current timeslot group g_{t_i};
4   Extract device list D in g_{t_i} based on User_Logs;
5   for d_1, d_2 ∈ D do
6     weight = W(d_1, d_2);
7     G.add_node(d_1,d_2);
8     G.add_edge(d_1,d_2,weight);
9   end
10   MN.append(G);
11 end
12 return MN
```

## Multilayer network analysis

Given a multilayer network, unsupervised graph analysis algorithms such as multilayer network embedding [39] and multilayer network community detection [28, 40] can be applied to extract useful information from the multilayer network automatically.

In this section, we discuss identifying abnormal nodes by analyzing network topology. To this end, we propose a basic assumption from the behavior analysis field: "Abnormal (markedly unusual) behavior is but a small fraction of all behavior of the current node [41]—most user behavior is normal," and we treat this as our baseline. With this assumption, identifying abnormal behavior for a user is equivalent to locating a small collection of nodes whose topologies are unlike the rest of the nodes in the corresponding multilayer network.

Below, we leverage multilayer network embedding [39], a state-of-the-art ML embedded topology-based analysis method, as well as multilayer network community detection [28], a purely topology-based analysis method to group nodes in the multilayer network, to locate small groups of nodes which share similar behavior.

**Multilayer network community detection.** By optimizing the multislice-modularity $Q$ [28] (Eq 2), this method groups nodes into different communities. Here, lowercase letters ($i$, $j \in N$) represent nodes; uppercase letters ($S, R \in L$) represent layers; $W_{ijS}$ is the weight between nodes $i$ and $j$; $Wk_{iS}$ is the total weight of node $i$, that is, the sum of all the weights between node

$i$ and its neighbors in layer $S$; $2\mu = \sum_{jR} k_{jR}$; $Wm_S = \sum_j Wk_{jS}$; $\gamma_S$ is the resolution parameter for layer $S$ [42]; $C_{jSR}$ indicates whether the node $j$ exists in both layes $S$ and $R$; $g_{iS}$ indicates whether the group of nodes $i$ exists in layer $S$; and $\delta_{A,B}$ is the identity function, where $\delta = 1$ when $A = B$.

$$Q = \frac{1}{2\mu} \sum_{ijSR,j,r \in N; S, R \in L} \left[ \left( W_{ijS} - \gamma_S \frac{Wk_{iS} Wk_{jS}}{2Wm_S} \delta_{SR} \right) + \delta_{ij} C_{jSR} \right] \delta_{g_{iS}, g_{jR}}. \tag{2}$$

Alg. 3 details the detection of communities in a multilayer network by maximizing $Q$, where the inputs are the multilayer network $MN$ and the resolution parameter $\gamma$ and the output is the community results $Coms = [com1, com2, \ldots]$ and $com = [d1, d2, \ldots]$. $neighbor(i, R)$ indicates the neighbors of node $i$ in layer $R$.

**Algorithm 3**: Weighted Multilayer Network Community Detection

```
Input: MN
Output: Coms
1 Coms = [];
2 stopFlag = False, Q = 0;
3 while !stopFlag do
4   StableComFlag = False;
5   while !StableComFlag do
6     MaxQ = Q;
7     for i, R ∈ MN do
8       Merge nodes i and j ∈ neighbor(i, R) into one Com by maximizing
Q;
9       Update MaxQ;
10       if Coms becomes stable then
11         Save Coms corresponding to MaxQ; Q = MaxQ;
12         StableComFlag = True;
13       end
14     end
15     Shrink each com ∈ Coms to one node, and update origin MN;
16     // Stop when they are shrunk to one node;
17     if len(Coms) == 1 then
18       stopFlag = True;
19     end
20   end
21 end
22 return Coms
```

**Multilayer network embedding.** This method uses the hyper-parameters $p$, $q$, and $r$ to enable a second-order random walk. Parameters $p$ and $q$ control the local and global biases of the sample random walk, and $r$ controls the layer traversals during the random walk; that is, the random walk stays on the current layer $l'$ with probability $r$, and moves along the edge of another layer $l$ with probability $1 - r$. Eq 3 gives the random walk traversal probability $P(t_i = (x^l, y^l, l)|t_{i-1} = (z^{l'}, x^{l'}, l'))$ among layers, where $(z^{l'}, x^{l'}, l')$ represents the first random walk from node $z^{l'}$ to node $x^{l'}$ in layer $l'$, and $(x^l, y^l, l)$ represents the second random walk from node $x^l$ to node $y^l$ in layer $l$. Note that $x^{l'}$ and $x^l$ is the same name node $x$ exists in different layers.

$$P(t_i = (x^l, y^l, l)|t_{i-1} = (z^{l'}, x^{l'}, l'))$$
$$\propto \begin{cases} \alpha_{pq}(z, x, l) r & \text{if} \quad l = l' \\ \dfrac{\alpha_{pq}(z, x, l)}{x_{|L|} - 1}(1 - r) & \text{otherwise.} \end{cases} \tag{3}$$

Here, analogous to $p$ and $q$ in node2vec [34], $\alpha_{pq}(z, x, l)$ indicates the traversal probability in the same layer $l$, given in Eq 4. $d_{zx}^l$ is the shortest path between nodes $z$ and $x$ in layer $l$ of the

multilayer graph (where nodes $z$ and $x$ may be the same node).

$$\alpha_{pq}(z, x, l) = \begin{cases} 1/p & \text{if} \quad d_{zx}^l = 0 \\ 1 & \text{if} \quad d_{zx}^l = 1 \\ 1/q & \text{if} \quad d_{zx}^l = 2 \\ 0 & \text{otherwise.} \end{cases} \tag{4}$$

Alg. 4 details the identification of the projection function $f$ that embeds the multilayer network into a vector space (Please see [39] for detailed information on Alg. 4), where *node2-vecSGD* corresponds to the running of stochastic gradient descent on minimizing negative node2vec log-likelihood with multilayer random walks taking the place of the standard node2-vec walks.

**Algorithm 4**: node2vec on Multilayer Network

```
Input: MN, r, α_pq, num_walks, walk_length
1 Initialize walk_list to empty;
2 for nw_iter from 1 to num_walks do
3   Initialize current edge (i, j, l) ← (i₀, j₀, l₀) uniformly at
random;
4   for wl_iter from 1 to walk_length do
5     walk_list[nw_iter][wl_iter] ← i;
6     With probability r, choose next_layer = l, otherwise choose nex-
t_layer = l' uniformly at random for some layer l' incident to j;
7     Set current edge (i, j, l) ← (j, i', next_layer) proportional to
α_pq(j, i', next_layer) for some i' incident to j through next_layer;
8   end
9 end
10 f ← node2vecSGD(walk_list)
```

## Outlier recommendation

Generally speaking, the community detection algorithm yields the subgroups of nodes that share similar behavior; thus the focus is on depicting relationships among nodes and their nearby nodes. The network embedding algorithm, in contrast, assigns nodes with coordinates on a vector space, making it suitable to apply a clustering algorithm (such as the K-means algorithm) on these nodes.

Alg. 5 combines the results of community detection along with the clustering results in vector space, calculates the score on each node, and recommends nodes with smaller scores.

**Algorithm 5**: Outlier Recommendation

```
Input: MN, Coms, Coordinates
Output: node_list
1 ComScores = [];
2 CoordinateScores = [];
3 Clusters ← clustering Coordinates into len(Coms) clusters;
4 foreach com ∈ Coms do
5   score = ∑_{i,j∈com} W(i, j);
6   ComScores.append(score);
7 end
8 foreach cluster ∈ Clusters do
9   score = ∑_{i,j∈cluster} W(i, j);
10   CoordinateScores.append(score);
11 end
12 foreach n ∈ fN do
```

```
13   Calculate node_score from ComScores and CoordinateScores;
14    node_list.append(node_score);
15 end
16 return node_list;
```

## Visualization system architecture

Fig 4 illustrates the architecture of the visualization system, showing the different tasks for a specific user.

- **User statistics extraction**: This task extracts all information related to the user. That is, by analyzing the user logs for each timestamp, this task obtains all the nodes in the multilayer network, calculates the time intervals, and gets the related timeslot groups for the current user.

- **Multilayer network visualization**: This task constructs the multilayer network for the related user. Here, each color represents the same name node (linked by a dotted line across different layers) in the network (Take Fig 5 in Section 16 as an example).

- **Abnormal behavior recommendation**: If there is any abnormal nodes in the current multilayer network, this task recommends them. At the same time, it also shows the group score for devices (We group devices together if they share the same score), calculates the security score for the user, and provides a report for current suspicious activities. Here, the security score is defined as $s = 1 - \frac{|abnormal\_nodes|}{|all\_nodes|}$.

## Experiments

We first describe the experimental settings (Section 16) and data sets (Section 16). Then we conduct a detailed evaluation of our proposed system from five aspects: in Section 16, we compare the advantages and disadvantages of the proposed system and the existing methods; in Section 16, by injecting abnormal devices, we evaluate the system performance in terms of precision rate, recall rate, and F1-score; in Section 16, from the view of the system operator, we present the time complexity for each proposed algorithm, and show the system response time based on the whole dataset; in Section 16, we demonstrate our private-preserving

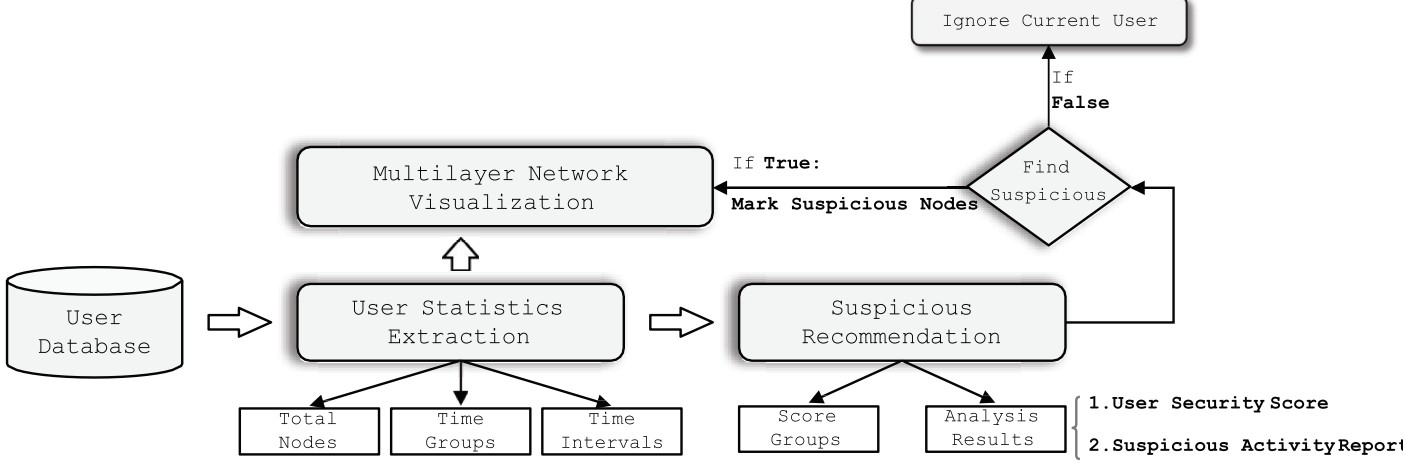

**Fig 4. Architecture of visualization system.**

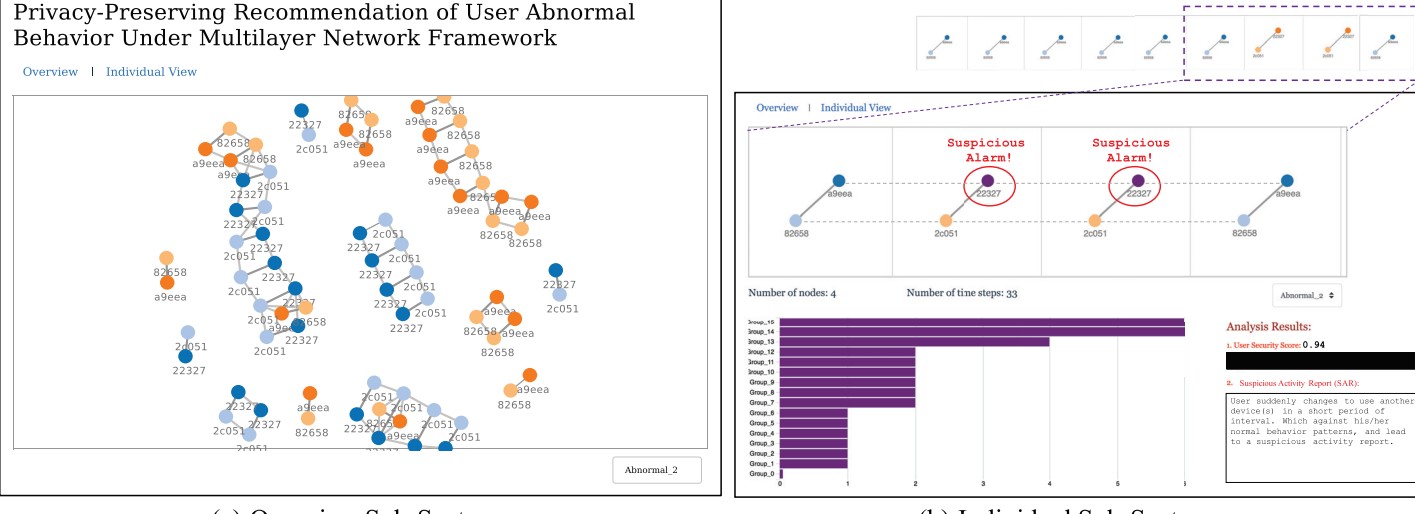

**Fig 5. Visualization system.** Proposed privacy-preserving recommendation system for abnormal user behavior. System has two major parts: an overview (left), and individual behavior (right) along with device score groups, analysis results, and suggestions.

recommendation visualization system; and in Section 16, we present user study from "Wen-juanxing" crowd-source system and "Amazon Mechanical Turk".

## Experimental settings

To demonstrate the proposed privacy-preserving abnormal user behavior recommendation system, we implemented the proposed algorithms as a prototype system on an Ubuntu 16.04 LTS based workstation with an Intel Xeon E5-2630 v4 processor and 64GB of 2133MHz RAM. We used this workstation for evaluating our system security, recommendation accuracy, and system performance.

## Data sets

In our experiments, we use the famously posted (2017.06.29) Tianchi dataset OneID (https://tianchi.aliyun.com/datalab/dataSet.htm?spm=5176.100073.888.2o7.40bc1022nJtK6m&id=20) to numerically evaluate the performance of the proposed system in Sections 16, 16 and 16. Table 1 shows some statistics for the OneID dataset.

**Table 1. Evaluation dataset.**

| Entire Users Information | | |
| --- | --- | --- |
| Users | Devices | Encrypted Search keywords and IPs |
| 453,029 | 926,578 | 43,786,036 |
| Per User Information | | |
| Avg timeslot groups | Avg devices | Avg Encrypted search keywords and IPs |
| 5.263 | 2.05 | 96.652 |

Notes: Despite a large number of logs, there are on average too few encrypted features per user to train good models. We use Alg. 1 to group times from each user's reach_time and data_time logs.

This dataset provides the encrypted users' wireless and PC browser logs for one week (2017.05.01–2017.05.07), with log information for 453,029 users, and 926,578 related devices. After cleaning the dataset, we found more than 40 million encrypted search keywords and scrambled IP addresses that can be used to depict user behavior. However, note that despite the large volume of encrypted features from the dataset, the average number of features for each user is limited, which complicates the training of user-specific models to capture corresponding behavior using ML-based algorithms.

## Method comparisons

Here, we compare our proposed system with three types of state-of-the-art privacy-preserving algorithms: a) robust deep autoencoders [16], b) ML based algorithms (linear regression, logistic regression, decision trees, random forests, support vector machines, and multi-layer perceptrons) [5], and c). asynchronous multi-task learning [7]. Table 2 summarizes the features for each type of algorithm. The proposed method has the following benefits:

1. **Preserves privacy**: As the construction of the multilayer network does not necessitate the use of feature details, the method naturally preserves user privacy at all times.

2. **Supports limited input features**: The multilayer network in our proposed method can be constructed via only a few (encrypted) features; however, the more (encrypted) features, the more accurate the resultant network.

3. **Is unsupervised**: Network embedding and community detection directly reveal subgroups with similar behavior in the multilayer network. Since in larger groups most nodes share similar behavior, it is reasonable to recommend those nodes in small groups as abnormally-behaving nodes.

4. **No need to train**: By leveraging multilayer network analysis, we do not need first to train a model. In contrast, the proposed algorithm obtains the results cognitively; this saves time compared to ML-based algorithms.

5. **Analyzes different sources**: As each layer in the multilayer network captures device behavior for the related timeslot group, different sources reveal the behavior of different devices of the specific user.

6. **Is noise-resistant**: Multilayer network analysis enhances signal-noise separation during the analysis process [8]; the proposed method inherits this advantage.

**Table 2. Comparison with state-of-the-art algorithms.**

| Related properties | Proposed method | Robust deep auto encoder | ML-based algorithms | Asynchronous MTL |
|---|---|---|---|---|
| 1. Privacy-preserving | Embedded | Required | Embedded | Embedded |
| 2. Volume of input features | Few | Large | Large | Large |
| 3. Unsupervised | Yes | Yes | No | No |
| 4. Training-free | Yes | No | No | No |
| 5. Analyzes different sources | Yes | No | No | Yes |
| 6. Noise-resistance | Not Effected | High | Not Effected | High |

Notes: ML = machine learning, MTL = multi-task learning

**Table 3. Abnormal device detection.**

| Injected devices | Precision | Recall | F1-score |
|---|---|---|---|
| 1 | 0.901 | 1.0 | 0.934 |
| 2 | 0.896 | 1.0 | 0.931 |
| 3 | 0.913 | 1.0 | 0.942 |

Notes: The high precision rate demonstrates the proposed system's high accuracy in distinguishing injected abnormal devices from other normal devices. The consistent, perfect recall rate shows that the system captures all injected abnormal devices.

## Abnormal device detection

We expect the proposed system to prove especially useful in scenarios where user logs reveal abnormal user behavior, for instance when the behavior of the current device is completely different from the behavior of the other devices.

Since there is no ground truth per user, we inject synthetic suspicious device logs into normal logs for a user, construct the multilayer network based on these logs, and use precision rate, recall rate, and F1-score [43] to evaluate the performance of the system. Here, for synthetic abnormal devices, the logged features such as encrypted search keyword and scrambled IP are totally different from other devices on the current timestamp; that is, the injected abnormal devices are isolated nodes in a layer.

After randomly injecting synthetic abnormal device logs into normal logs for 1000 users, Table 3 shows the average precision rate, recall rate, and F1-score for the system in scenarios where one to three abnormal devices are injected, respectively.

## System efficiency evaluation

Note that the proposed system is designed to maintain account security for a specific user, without violating user privacy. From the system operator's point of view, this is equivalent to establishing a privacy-preserving system for each user. However, since each account is independent, it is easy for operators to use a distributed and parallel approach to speed up the setup of the proposed system.

Here, we give a detailed description on time complexity analysis for all algorithms.

- The time complexity for constructing the proposed multilayer network for a user using Alg. 1 is $O(T)$, where $T$ is the total number of timestamps for the user.

- The time complexity of Alg. 2 is $O(\sum_t D_t^2)$, where $D_t$ is the total number of devices in the current timeslot group $t$.

- Secondly, the time complexity of Alg. 3 is $O(n \log n)$, where $n$ is the number of nodes in the multilayer network corresponding to the user.

- The time complexity of Alg. 4 is the linear $O(n)$ [34], and that of Alg. 5 is the linear $O(Com + Cluster + n)$, where $Com$ and $Cluster$ represent the number of communities and clusters respectively.

In addition, we also evaluate the efficiency for the proposed system based on the OneID dataset. Table 4 shows the consuming times for multilayer network construction and recommendation, respectively.

**Table 4. System efficiency.**

|  | Min T(ms) | Avg T (ms) | Max T (ms) |
|---|---|---|---|
| Step 1 Construction | 0.19 | 5.85 | 17.51 |
| Step 2 Recommendation | 0.61 | 2.23 | 4.45 |

<u>Notes</u>: The overall time to system construction and recommendation steps is around 10ms; from the system operator's point of view, this indicates the proposed system is fast enough for large-scale deployment [1].

## Visualization system

Fig 5 shows the user interface for the proposed system. Here, we use the OneID dataset to introduce the system. By randomly selecting a user from the database, the Overview subsystem quickly builds the topology of all the devices associated with the user. Then the Individual subsystem quickly constructs a multilayer network composed of these devices, and based on the above algorithm, the subsystem automatically calculates the current user's user security score, recommends the user's abnormal device(s), and supplies the reason for its judgment.

Here, we randomly choose one account (Abnormal_2) as an example to help readers understand how our system works: From the Overview subsystem, this visualization system shows the topological information for all devices related to the current user. Different colors here represent different users. In the individual subsystem, it automatically marks <u>Device 22327</u> as abnormal node in the multilayer network. Below the multilayer network diagram, the system extracts the user statistics (with 4 devices and 33 time steps) related to the current node (Abnormal_2), after which the group information shows nodes with like scores (clustered into 16 groups), where Device 22327 is located in the first group with the smallest score ($\simeq 0$). From the Overview subsystem, it is difficult to distinguish that these are abnormal devices. From the Individual subsystem, however, we scroll the multilayer network view horizontally to see why this device is abnormal. Here, we attach the re-scaled longer time window (up to 9 layers) upon the individual subsystem diagram, revealing that 22327 is not a major device for these timestamps. Hence, our system easily reasons that 22327 is an abnormal device for these timestamps, as users primarily use other devices at this time.

## User case study

To further verify the ease of use of the proposed visualization system, we designed a user questionnaire to evaluate our system from the perspective of performance, accuracy, and reasoning precision. We gathered 127 samples from the "Wenjuanxing" crowd-source system (https://www.wjx.cn/) and 218 samples from "Amazon Mechanical Turk" (https://requester.mturk.com/).

First, Table 5 provides details on the questionnaire. Here, we use Cronbach's $\alpha$ coefficient [44] to evaluate the reliability of our questionnaire. In general, Cronbach's $\alpha$ coefficient is 0.832, larger than the minimum requirement (0.7), which gives us confidence that the designed questionnaire is reliable. Shown in Fig 6 is the average score for each question, where the horizontal axis represents the score from 0 to 1 and the vertical axis represents the nine questions. Overall, the result statistics show that almost all feedbacks have achieved top 15% satisfaction. Below we give a detailed analysis on each results:

- **Overall system (Q1, Q2)**: The average score of the overall system is about 0.8, indicating that the proposed visualization system performs well in terms of ease of use and system performance;

**Table 5. System efficiency.**

| No. | Question | Score |
|---|---|---|
| Q1 | The overall response speed of the system | 1–5* |
| Q2 | The user-friendliness of the system | 1–5* |
| Q3 | Does "Overview system" show all the device information for the current user? | Yes/no** |
| Q4 | Does "Overview system" show the topological behavior among all related devices? | Yes/no** |
| Q5 | Does "Individual system" represent the device behavior within a certain timeslot group? | Yes/no** |
| Q6 | Does "Individual system" capture the topological relationship within a certain timeslot group? | Yes/no** |
| Q7 | Is it possible to clearly identify abnormal nodes in "Individual system"? | Yes/no** |
| Q8 | Does "User security score" represent the security status for the related user? | Yes/no** |
| Q9 | Does "Suspicious activity reports" identify abnormal devices convincingly enough? | 1–5* |

Notes: Cronbach's $\alpha$ coefficient is 0.837.

*: 1–5 indicates users choose a score from 1 to 5, where 1 is the worst and 5 the best.

**: For Yes/no questions, we use a score of 1 for yes and 0 for no when calculating the average score for related questions.

- **Overview subsystem (Q3, Q4)**: The overall score is greater than 0.9, demonstrating that the subsystem displays information about all devices associated with the current user;

- **Individual subsystem (Q5–Q9)**: Other than question Q7, all scores indicate that the "Multi-layer Network Construction," "Devices Score Groups," and "Analysis Results Reasoning" modules recommend the anomaly for the current user. For Q7, we believe this is due to insufficient domain knowledge in multilayer network analysis. In the future, we will improve

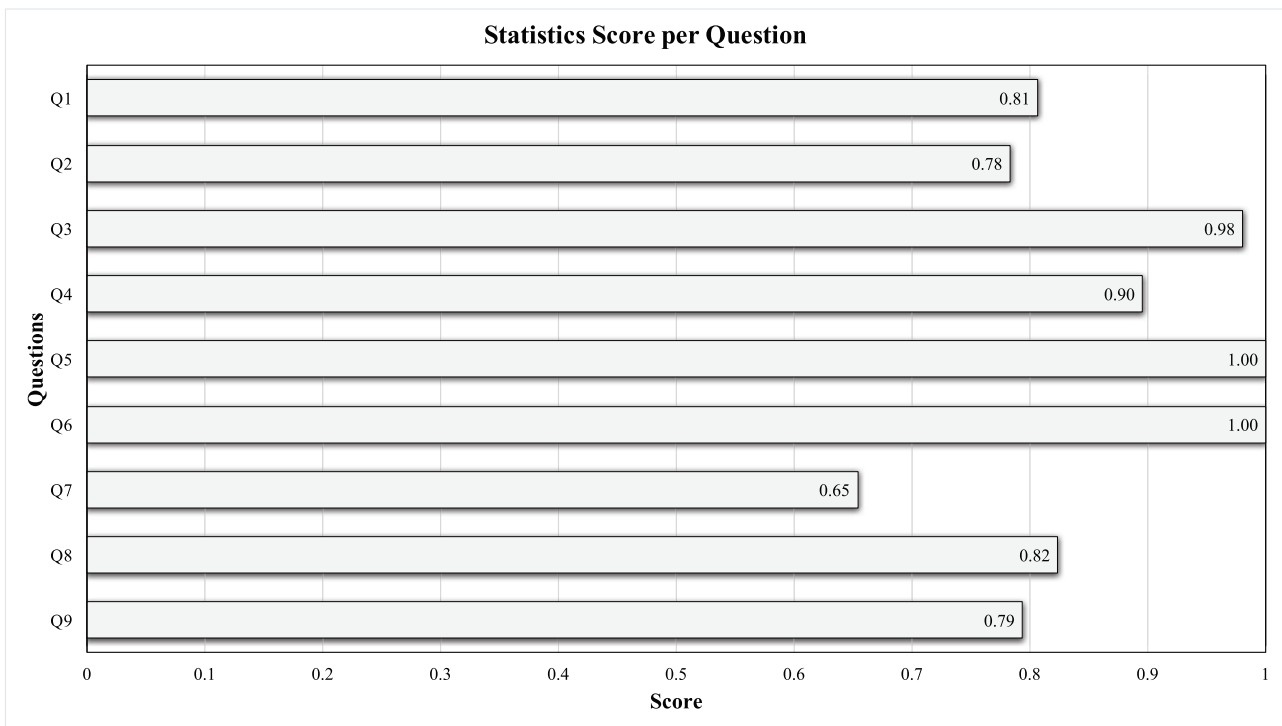

**Fig 6. Statistic results for user case study.**

the visualization method to better reflect abnormal behavior so as to enhance the overall system.

## Conclusion

In this paper, we propose a privacy-preserving recommendation system under a multilayer network framework; the system automatically recommends abnormal user behavior without compromising the privacy of the corresponding user. In our system, the graph-based construction module and analysis module are what make recommendations work in practice: The former embeds user behavior across time as the topological features of a multilayer network, and the latter incorporates the analysis results from a pure topology-based method and the state-of-the-art multilayer network embedding method, and detects and recommends user abnormal behavior within the multilayer network, without pre-training a model in advance. Our experiments show that the system is accurate and efficient, with the F1-score greater than 0.93, and the overall system response time less than 22ms. In addition, from the system operator's point of view, we find that our system is efficient enough to be deployed for each user in an online service. To prove this, we also conduct an experiment deploying the system for each user in the OneID dataset, which includes more than 40 thousand nodes and 43 million encrypted features. In addition, the user-case results from "Wenjuanxing" crowd-source system and "Amazon Mechanical Turk" show that our proposed practical system is highly friendly to user, with the overall feedbacks up to 85% satisfaction.

## Acknowledgments

We thank the Alibaba company for providing free and public data sets. We also thank the reviewers and editors for their valuable comments and construction criticism.

## Author Contributions

**Conceptualization:** Chengyun Song, Weiyi Liu.

**Data curation:** Chengyun Song.

**Formal analysis:** Chengyun Song.

**Funding acquisition:** Chengyun Song.

**Methodology:** Chengyun Song, Weiyi Liu.

**Project administration:** Zhining Liu.

**Resources:** Chengyun Song, Xiaoyang Liu.

**Software:** Weiyi Liu, Zhining Liu.

**Supervision:** Zhining Liu.

**Validation:** Zhining Liu.

**Visualization:** Weiyi Liu, Zhining Liu.

**Writing – original draft:** Zhining Liu.

**Writing – review & editing:** Zhining Liu, Xiaoyang Liu.

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
