## [Decision Letter · Decision Letter 0]

4 Sep 2019

PONE-D-19-18509

Privacy-Preserving Recommendation of User Abnormal Behavior Under Multilayer Network

PLOS ONE

Dear Dr. song,

Thank you for submitting your manuscript to PLOS ONE. After careful consideration, we feel that it has merit but does not fully meet PLOS ONE’s publication criteria as it currently stands. Therefore, we invite you to submit a revised version of the manuscript that addresses the points raised during the review process.

We would appreciate receiving your revised manuscript by Oct 19 2019 11:59PM. To enhance the reproducibility of your results, we recommend that if applicable you deposit your laboratory protocols in protocols.io, where a protocol can be assigned its own identifier (DOI) such that it can be cited independently in the future. For instructions see: http://journals.plos.org/plosone/s/submission-guidelines#loc-laboratory-protocols

We look forward to receiving your revised manuscript.

Kind regards,

He Debiao

Academic Editor

PLOS ONE

Journal Requirements:

'The authors have declared that no competing interests exist'

We note that one or more of the authors are employed by a commercial company: JD Urban Computing Business Unit.

Additional Editor Comments (if provided):

Reviewers' comments:

Reviewer's Responses to Questions

**Comments to the Author**

1. Is the manuscript technically sound, and do the data support the conclusions?

Reviewer #1: Yes

Reviewer #2: Yes

2. Has the statistical analysis been performed appropriately and rigorously? 

Reviewer #1: Yes

Reviewer #2: Yes

3. Have the authors made all data underlying the findings in their manuscript fully available?

Reviewer #1: Yes

Reviewer #2: Yes

4. Is the manuscript presented in an intelligible fashion and written in standard English?

Reviewer #1: Yes

Reviewer #2: Yes

5. Review Comments to the Author

Reviewer #1: I am particularly satisfied with content and structure of this article. It suggest that the author has taken enough time to prepare the manuscript. The few observations from me is highlighted in the article.

Reviewer #2: The manuscript presents important information and deserves to be published in Plos One. Before publication, I suggest the following modifications:

Page 1, line 6-7

Change “machine learning algorithms” for “machine learning (ML) algorithms”

Page 1, line 7

Change “SVM” for “Support Vector Machine (SVM)”

Page 1, line 9

Change “machine learning algorithms” for “ML algorithms”

Page 1, line 13

Change “IP adresses” for “Internet Protocol (IP) addresses”

Page 1, line 15

Change “machine learning models” for “ML models”

Page 2, line 18

Change “machine learning methods” for “ML methods”

Page 2, line 27

Change “machine learning algorithms” for “ML algorithms”

Page 2, line 30

“…over time1.”

Confirm in the Plos One standards if footnotes can be used.

Page 2, line 39

Change “machine learning” for “ML”

Page 2, line 55

“The system is available at https://github.com/StoneSongLucky/Private Preserving Outlier Behavior Detection”

The link is unavailable.

Page 2, line 56-62

Delete: “The remainder of the paper is organized as follows. In Section 2, we discuss recent work on user abnormal behavior recommendation. Section 3 describes the whole proposed model, and describes network construction (Section 3.3), network analysis (Section 3.4), outlier recommendation (Section 3.5) and a visualization practical system (Section 3.6). In Section 4, we present a thorough evaluation of the proposed privacy-preserving recommendation system. Finally, we conclude and describe future work in Section 5.”

Page 3, line 65

Change “machine learning” for “ML”

Page 3, line 71

Change “machine learning (ML) models” for “ML models” - The abbreviation should appear the first time the word is cited in the manuscript (as per the previous changes I am suggesting).

Page 10, line 262

Change “machine learning-based algorithms” for “ML-based algorithms”

Page 10, line 266

Change “machine learning-based algorithms” for “ML-based algorithms”

Page 11, line 283

Change “machine learning-based algorithms” for “ML-based algorithms”

6. PLOS authors have the option to publish the peer review history of their article (what does this mean?). If published, this will include your full peer review and any attached files.

Reviewer #1: Yes: MKA Abdulrahman

Reviewer #2: No

---

## [Author Response · Author response to Decision Letter 0]

8 Oct 2019

Response to Reviewers and Editors' Comments 

& List of Changes in the Revised Paper

Title: Privacy-Preserving Recommendation of User Abnormal Behavior Under Multilayer Network

Authors: Chengyun Song, Weiyi Liu, Zhining Liu, Xiaoyang Liu

The authors thank the reviewers and editors for their comments and suggestions. We have revised the paper following the comments and suggestions. Following is our response and also the list of changes that we have made in the revised version. The changes, including newly added parts, are in red font in the revised manuscript.

Review Comments to Author:

Reviewer 1:

1. Reconcile the 15% and 85% satisfaction mentioned in the conclusion

Response:

Sorry to bring some mistakes and ambiguous here. We meant to say that according to our proposed practical system is highly friendly to uses, as we use “85% satisfaction” to point out there are a large amount of users that do accept our proposed system, and only “15%” of users which are NOT satisfied. To eliminate the ambiguous, we use “up to 85% satisfaction” in the abstract and conclusion

2. Citation missing

Response:

In the paper, we define a simple and way (the percentages of abnormal nodes) to evaluate the security scores in multilayer network. So, there is no citation here.

3. From the decision box " If True". The flow chart did not say anything about 'if false". Marking the suspicious node is not enough but creating a library of suspicious nodes to be verify for subsequent logins

Response:

For “If false”, the program will ignore the current user. We have updated this figure in the revised paper. And if the system has found a “suspicious” node, it will raise an alert to alarm the user, as the final decision should be made by user himself.

4. Give the scenario of medium level noise in this method

Response:

Here we use “medium” as the opposite of “High”. In fact, due to the fact that our proposed method (privacy-preserving) and the ML-based algorithms needs all the information to learn the user behavior, these methods have possibilities to absorb noises information as well. Hence, we have replaced term “medium” as “Not Effected”.

5. Cite standard that support this assertion

Response:

 We have added the cite paper to support the assertion in the revised version.

6. Reconcile with abstract

Response:

Sorry to bring ambiguous here. We have used “up to 85% satisfaction” in the abstract which is same to the conclusion.

Reviewer 2:

1. Page 1, line 6-7

Change “machine learning algorithms” for “machine learning (ML) algorithms”

Response:

I have added the abbreviation in the revision (Page 1 line 6-7).

2. Page 1, line 7

Change “SVM” for “Support Vector Machine (SVM)”

Response:

I have added the abbreviation of “SVM” in the revision (Page 1 line 7).

3. Page 1, line 9

Change “machine learning algorithms” for “ML algorithms”

Response:

I have used the abbreviation in the revision (Page 1 line 9).

4. Page 1, line 13

Change “IP adresses” for “Internet Protocol (IP) addresses”

Response:

I have given the full name of IP in the revision (Page 1 line 14).

5. Page 1, line 15

Change “machine learning models” for “ML models”

Response:

I have used the abbreviation of “ML” in the revision (Page 2 line 15).

6. Page 2, line 18

Change “machine learning methods” for “ML methods”

Response:

I have used the abbreviation of “ML” in the revision (Page 12 line 18).

7. Page 2, line 27

Change “machine learning algorithms” for “ML algorithms”

Response:

I have used the abbreviation of “ML” in the revision (Page 2 line 27).

8. Page 2, line 30

“…over time1.” Confirm in the Plos One standards if footnotes can be used.

Response:

We have not found the standards about the footnotes. However, this standard is used in other journals. So I do no changes and keep the original format in the revision (Page 2 line 30).

9. Page 2, line 39

Change “machine learning” for “ML”

Response:

I have used the abbreviation of “ML” in the revision (Page 2 line 39).

10. Page 2, line 55

“The system is available at https://github.com/StoneSongLucky/Private Preserving Outlier Behavior Detection” The link is unavailable.

Response:

The correct links are 

https://github.com/StoneSongLucky/Private_Preserving_Outlier_Behavior_Detection

and 

https://github.com/Liu-WeiYi/Private_Preserving_Outlier_Behavior_Detection

The link in the paper is right and available.

11. Page 2, line 56-62

Delete: “The remainder of the paper is organized as follows. In Section 2, we discuss recent work on user abnormal behavior recommendation. Section 3 describes the whole proposed model, and describes network construction (Section 3.3), network analysis (Section 3.4), outlier recommendation (Section 3.5) and a visualization practical system (Section 3.6). In Section 4, we present a thorough evaluation of the proposed privacy-preserving recommendation system. Finally, we conclude and describe future work in Section 5.”

Response:

I have deleted those contents in the revision.

12. Page 3, line 65

Change “machine learning” for “ML”

Response:

I have used the abbreviation of “ML” to replace ‘machine learning’’ in the revision (Page 2 line 57).

13. Page 3, line 71

Change “machine learning (ML) models” for “ML models” - The abbreviation should appear the first time the word is cited in the manuscript (as per the previous changes I am suggesting).

Response:

I have revised those mistakes in the revision (Page 3 line 64).

14. Page 10, line 262

Change “machine learning-based algorithms” for “ML-based algorithms”

Response:

I have used the abbreviation of “ML” in the revision (Page 10 line 254).

15. Page 10, line 266

Change “machine learning-based algorithms” for “ML-based algorithms”

Response:

I have used the abbreviation of “ML” in the revision (Page 10 line 257).

16. Page 11, line 283

Change “machine learning-based algorithms” for “ML-based algorithms”

Response:

I have used the abbreviation of “ML” in the revision (Page 11 line 274).

---

## [Decision Letter · Decision Letter 1]

21 Oct 2019

Privacy-Preserving Recommendation of User Abnormal Behavior Under Multilayer Network

PONE-D-19-18509R1

Dear Dr. liu,

We are pleased to inform you that your manuscript has been judged scientifically suitable for publication and will be formally accepted for publication once it complies with all outstanding technical requirements.

With kind regards,

He Debiao

Academic Editor

PLOS ONE

Additional Editor Comments (optional):

Reviewers' comments:

Reviewer's Responses to Questions

**Comments to the Author**

1. If the authors have adequately addressed your comments raised in a previous round of review and you feel that this manuscript is now acceptable for publication, you may indicate that here to bypass the “Comments to the Author” section, enter your conflict of interest statement in the “Confidential to Editor” section, and submit your "Accept" recommendation.

Reviewer #1: All comments have been addressed

Reviewer #2: All comments have been addressed

2. Is the manuscript technically sound, and do the data support the conclusions?

Reviewer #1: Yes

Reviewer #2: Yes

3. Has the statistical analysis been performed appropriately and rigorously? 

Reviewer #1: I Don't Know

Reviewer #2: Yes

4. Have the authors made all data underlying the findings in their manuscript fully available?

Reviewer #1: Yes

Reviewer #2: Yes

5. Is the manuscript presented in an intelligible fashion and written in standard English?

Reviewer #1: Yes

Reviewer #2: Yes

6. Review Comments to the Author

Reviewer #1: Better in the current form since all questions and comment has been addressed. Also, to the best of my understanding comply with Plos One policy. From basic research writing and reporting, the paper has improved significantly. Assessment on the technical aspect is left for other reviewers and the editorial board.

Reviewer #2: After the modifications, consider the appropriate manuscript for publication in Plos One in its current form.

7. PLOS authors have the option to publish the peer review history of their article (what does this mean?). If published, this will include your full peer review and any attached files.

Reviewer #1: Yes: MKA Abdulrahman

Reviewer #2: No

---

## [Editor Report · Acceptance letter]

13 Nov 2019

PONE-D-19-18509R1 

Privacy-Preserving Recommendation of User Abnormal Behavior Under Multilayer Network 

Dear Dr. Liu:

I am pleased to inform you that your manuscript has been deemed suitable for publication in PLOS ONE. Congratulations! Your manuscript is now with our production department. 

With kind regards,

on behalf of

Dr. He Debiao 

Academic Editor

PLOS ONE